# Valuation of Visitor Perception of Urban Forest Ecosystem Services in Kuala Lumpur

Emylia Shakira Jamean [1] and Azlan Abas [1,2,*]

1   Center for Research in Development, Social and Environment, Faculty of Science, Social and Humanities
    Universiti Kebangsaan Malaysia, Bangi 43600, Selangor, Malaysia
2   Centre of Excellence for Social Innovation & Sustainability (CoESIS), Universiti Malaysia Perlis,
    Jalan Kangar-Alor Setar, Kangar 01000, Perlis, Malaysia
*   Correspondence: azlanabas@ukm.edu.my

**Abstract:** Urban forests play a vital role in maintaining the city ecological balance and providing ecosystem services to citizens. Ecosystem services lead to better quality of life, better environmental quality, and more sustainable urban growth. However, many emerging nations have often progressed at the price of lowering and sacrificing forest coverage, which has a negative impact on the benefits that the public receives from natural green spaces. As a result, the goals of this research were to ascertain visitor impressions of urban forests in Kuala Lumpur, to assess the value assigned by visitors to urban forests in Kuala Lumpur, and to investigate the elements that impact the willingness to pay. A questionnaire-based field study was conducted on a total of 254 respondents among Taman Tugu Urban Forest visitors, Kuala Lumpur. The results show that visitor perceptions of regulating services, cultural services, and supporting services were positive, with high-level score values of 4.74, ±0.40, 4.69, ±0.37, and 4.70, ±0.50. Furthermore, provisioning services were perceived to be of moderately high level (3.49, ±1.12), and visitor perceptions of urban forest amenities were positive, with high-level scores (4.39, ±0.53). Overall, this indicates that visitors had a very good perception of Taman Tugu Urban Forest ecosystem services. However, when we looked at the factors that contributed to the willingness to pay for the conservation of urban forests and entry, only the perception of the amenities provided at Taman Tugu had a significant relationship with the willingness to pay. A total of 79.1% of visitors expressed their willingness to pay, for forest conservation, an average payment value of MYR 51.32 per year, while 65% of visitors were willing to pay, as entry fees at urban forests, an average payment value of MYR 3.07 per person. It can be concluded that visitors had a positive perception of urban forests in Kuala Lumpur and were willing to contribute for conservation and entry fee purposes. We hope that the findings of this research contribute to a better understanding of urban forest ecosystem services in Kuala Lumpur and visitor perception. In addition, this study could also be useful to policy makers to formulate a specific policy focus on urban forests by comprehensively and holistically including the monetary value of the ecosystem services provided, considering public opinion and needs, and performing financial allocation for conserving and managing urban forests. This is to ensure that the urban sustainable development goals and smart city aims can be achieved.

**Keywords:** urban ecosystem; ecosystem management; cultural services; environmental management; sustainability

## 1. Introduction

Urban forests are the main resource that provides ecosystem services to urban residents [1], by providing various benefits, including economic, social, ecological, and environmental contributions, as well as ecosystem services [2]. Furthermore, they provide a variety of ecosystem goods and services that can improve the quality of life of urban residents [3] and exert recovery effects on human health [4], including alleviating depression and anxiety

levels, relieving stress, and attention recovery [5]. Urban forests can also provide significant ecosystem services, such as reducing urban heat [6], removing air pollution [7], reducing noise, and regulating the microclimate [8], and function as biological indicators in the urban area. For example, lichen is used as a bioindicator to monitor the air quality in the area, since it cannot live in polluted areas [9], and at the same time, it has been used as a biological indicator to indicate the quality of human life [10]. Furthermore, urban forests help to reduce the occurrence of water runoff and erosion in nearby areas [11] and can also become a habitat for various species of flora and fauna [12], which acts as a complement to the ecosystem chain in urban forests. In addition, urban forests also contribute to the increase in property value in the area [13–15]. On the other hand, some functions of urban forest ecosystems can also be harmful to humans, thus becoming ecosystem disservices [16]. Examples of ecosystem disservices are infrastructure damage caused by the roots of trees and diseases or injuries caused by plants or wildlife.

Ecosystem services can be defined as the benefits that people derive from nature [17], as well as the ecological characteristics, functions, or processes that directly or indirectly contribute to human well-being [18]. Ecosystem services are classified into four main categories, namely, provisioning services, such as food, raw materials, and water; regulating services, such as flood regulation, carbon sequestration, drought, and disease; cultural services, such as recreational, spiritual, and aesthetic functions; and supporting services, such as nutrient cycle, soil formation, and habitat [19,20]. The rationale behind the use of the concept of ecosystem services is mainly to show how the loss of biodiversity would directly affect the ecosystem functions that provide support to critical services related to human well-being. The concept of ecosystem services is anthropogenic, and so is the ecosystem service assessment process. Because of this, efforts to assign a value to ecosystem services, especially financial value, have been criticized for solely preserving ecosystems for the benefit of humans [21,22]. In this context, the concept of ecosystem disservices denotes the processes and functions that affect humans in negative ways, causing damage and costs [23].

The valuation of ecosystem services has also become challenging because of the intrinsic value that humans assign to ecosystems, besides the challenge of measuring the economic value of non-traded ecosystem services [24]. Due to absence of an efficient market for these resources, many economic valuation techniques have been developed by economists to evaluate non-marketed goods [25,26]. The basic meaning of social value is the value that the community is willing to pay or able to pay to obtain a service (willingness to pay—WTP) or the value that can be accepted (willingness to accept—WTA). WTP is a concept introduced to determine consumer preference to pay for the demand for goods and services [27]. The estimation of WTP plays a major role in defining the economic value of the conservation of the ecosystem, and it can be derived from three major valuation methods—CVM, Conjoint Analysis, and Choice Experiments [28]; each method has different ways for making assumptions, and each is dependent on the situation of the study [29]. According to Langford et al. [30], people are willing to pay when natural resource conservation provides benefits to the community and future generations, especially in socioeconomic aspects. The CVM is a straightforward, easy-to-understand, and very flexible technique. The most used approach is directly asking the respondent whether or not they would be willing to pay a certain amount of money for the level of the non-marketed good described. This method is widely used all over the world in various areas of economics, such as cultural economics [31], and transportation safety and economics [32], as well as environmental economics. The development of the CVM in Malaysia started more than a decade ago, and was first used by Nik Mustapha [33] for valuing outdoor recreational resources in urban parks; Jamal and Redzuan [34] and Alias et al. [35] studied wetland benefits, and Zaiton [36] focused on conservation benefits of National Park.

The accelerating urbanization has led to environmental deterioration, ecosystem disturbance, fragmentation, the shrinking in green areas, and the reduction in open spaces

for recreational purposes [37]. It has also changed ecological functions and processes, disrupted ecosystem services, and affected human well-being. Over the last 50 years, Malaysia has quickly become an urban nation, and among the most densely populated areas is Kuala Lumpur, with an annual population growth rate of 2.2% in 2021 [38]. The increase in population in urban areas in Malaysia requires urgent attention from all parties involved. Adequate and up-to-date information on urban forests, including the public perception of urban forests, the monetary value of urban forests, and their environmental services in urban settings, is essential to future urban planning. The study of the perception of urban forest ecosystem services has been very poorly conducted [39]. Ko & Son [39] also mentioned that information regarding the community perception of urban forests is lacking, and Willemse [40] also agreed that there is a lack of research on the visitor perception of green spaces in developing countries. This is important because countries that are undergoing rapid urbanization often degrade green space for other land uses. Additionally, there is still no comprehensive review that specifically addresses urban forestry research in Malaysia [41]. Therefore, proper studies need to be conducted to justify the importance of green space and specifically urban forests. This research attempted to (i) determine the visitor perceptions of urban forests, (ii) evaluate the value assigned by visitors to urban forests in Kuala Lumpur, and (iii) analyze the factors that affect the willingness to pay.

## 2. Research Methodology

### 2.1. Research Area

This study was conducted at Taman Tugu (3.1514° N, 101.6846° E), the urban forest of Kuala Lumpur (refer Figure 1). The selection of Taman Tugu was due to the fact that this park is a new recreation area with an urban forest concept that was opened to the public less than five years ago. No scientific study has ever been carried out in this area, especially related to visitor perception, biodiversity, and urban forest ecosystems. Taman Tugu is located on the north side of Taman Botani Perdana and is surrounded by National Monument in the southwest, Padang Merbok in the southeast, and Lanai Kijang Bank Negara in the east. The park has an area of 66.7 hectares and has been gazetted as a permanent green area in the city of Kuala Lumpur by the Ministry of Federal Territories (KWP) under Kuala Lumpur City Plan [42]. The establishment of Taman Tugu as a natural forest conservation and preservation project was launched on 4 September 2016. To date, in collaboration with Forestry Research Institute of Malaysia (FRIM), there are more than 1000 trees in the Taman Tugu site area that have been identified and marked for conservation purposes. This includes several local tree species, such as Jelutong, Tembusu, Pulai, and Gaharu. There are trees that have a diameter of over one meter and are estimated to be over 100 years old. The average age of trees is between 8 and 10 years, and there are 230 species of rainforest trees. There are also species categorized as endangered or critically endangered by International Union for Conservation of Nature (IUCN), such as the Keruing, Meranti, and Mersawa species. Furthermore, Taman Tugu provides facilities such as large parking space, public toilets, rest areas, and forest trails, and the management also organizes activities for visitors, such as educational programs, tree planting activities, zumba, yoga, and many more (Figure 2) [43].

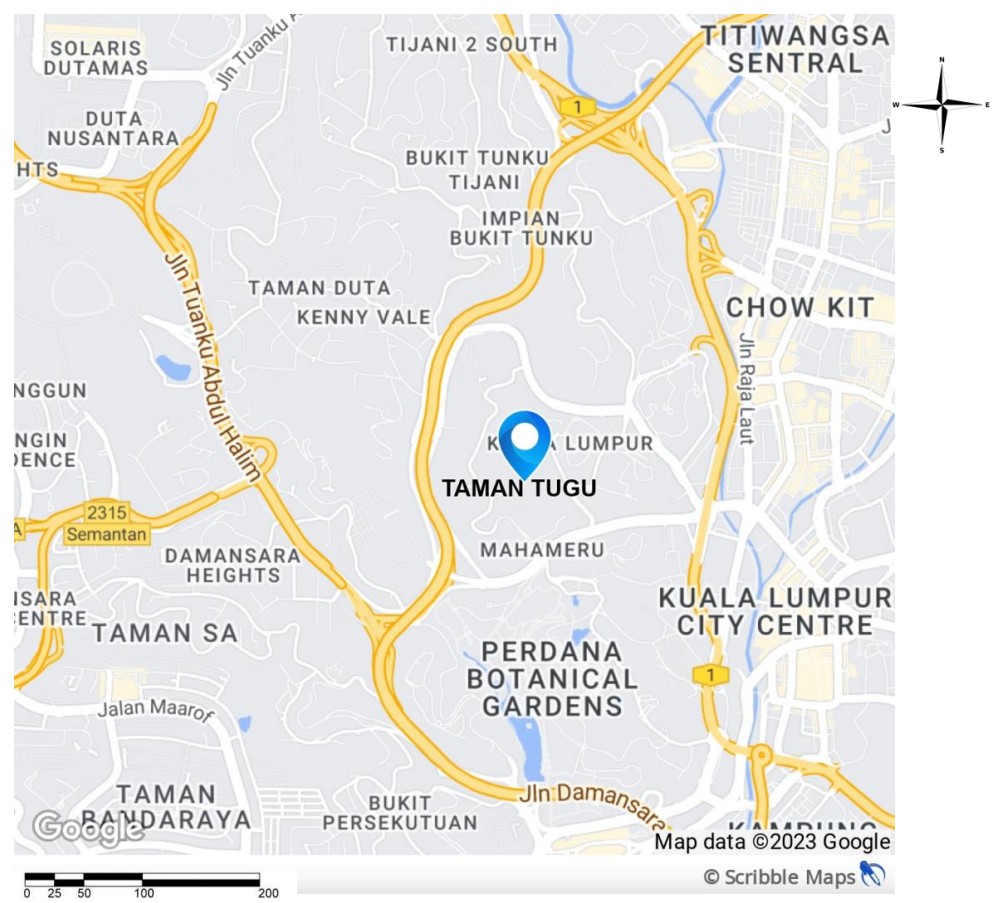

**Figure 1.** Location of Taman Tugu, Kuala Lumpur.

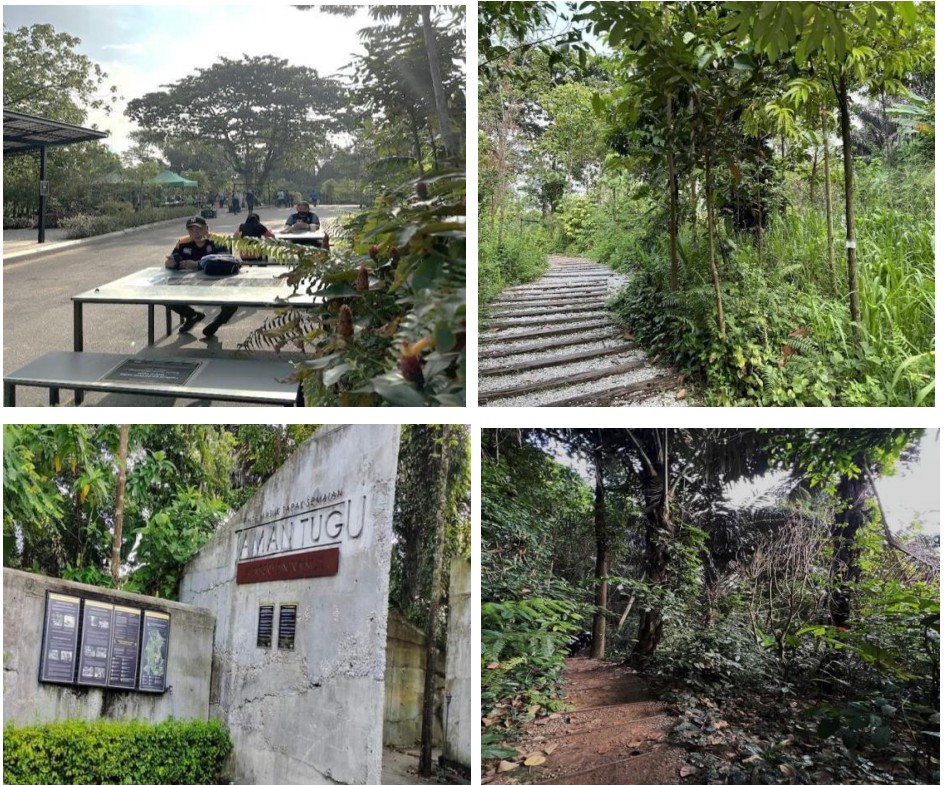

**Figure 2.** Photos of Taman Tugu (**left**), Kuala Lumpur (**right**).

## 2.2. Research Methods

This study was carried out using a quantitative approach, and the sampling technique used in this study was random sampling. The sample was randomly chosen among the visitors of the study area. A set of structured questionnaires was developed and used to collect information. Pre-testing was initially conducted before the actual survey to check the plausibility and ease of understanding of the instrument. The survey was conducted from January to February 2022, and the number of responses received from visitors randomly administered face-to-face questionnaires at Taman Tugu was 254. The questionnaires were adapted from Lagbas [44] and comprised five sections: Section one was about the demographic characteristics of respondents, which included age, gender, level of education, occupation, level of income, and other variables that may have had an influence on their WTP responses. The second section was designed to examine the respondents' knowledge of the terms related to the research, while the third and fourth sections were related to the visitor perception of the ecosystem services and the visitor perception of the amenities of urban forests, measured using a Likert scale. The last section was designed to determine the willingness to pay (WTP) for the conservation of urban forests and entrance of the respondents, as well as the amount that they were willing to pay for those purposes. To obtain answers that reflected the true maximum WTP of the respondents, a single-bounded, dichotomous choice was provided. The respondents were asked whether they were willing (yes/no answer) to pay any amount for the conservation of urban forests and urban forest entrance. If the respondents answered yes, they were asked to provide the amount that they were willing to contribute.

## 2.3. Research Analysis

A descriptive analysis that involved the use of tools such as frequency, mean, standard deviation, and percentage was performed to analyze the profile of respondents, such as age, race, level of education, and income. Descriptive statistics reveal the common type of response distribution of all variables. Specifically, mean and SD were used to identify the characteristics of the sample in this study. According to Hair et al. [45], mean values can be categorized into four levels: low, moderately low, moderately high, and high, as shown in Table 1.

**Table 1.** Mean values.

| Mean Score | Interpretation |
| --- | --- |
| 1.00–2.00 | Low |
| 2.01–3.00 | Moderately low |
| 3.01–4.00 | Moderately high |
| 4.01–5.00 | High |

Source: Hair et al. [45].

Findings from this descriptive analysis allowed the researchers to describe the variables involved as factors that influenced the level of knowledge of the respondents about the ecosystem and the perception of the respondents of ecosystem services, the facilities provided, and issues regarding urban forests. To test for possible factors that contributed to the visitor willingness-to-pay attitude, multiple regression analysis was performed. The SPSS v.28 software package was used for all statistical analyses.

## 2.4. Willingness to Pay

The approach used in the CVM is to ask the public to state their willingness to pay (WTP) to obtain a product, not to just draw conclusions based on observing behavior in a market context. There are four types of CVM methods, namely, the bidding game, payment card, open-ended choice, and dichotomous choice. For this study, the open-ended choice method was used. In this format, a respondent was asked two related questions: first, the respondent needed to state whether they would agree or not ('yes' or 'no') to spend

a certain amount of money for the conservation of urban forests; then, with the second question, the respondent needed to state whether they would agree or not with an entrance fee to urban forests being charged to visitors. If the answer was 'yes', they needed to state the amount that they would have been willing to contribute, and if the answer was 'no', they needed to state the reason. Next, the mean was calculated to determine the value given by visitors to the conservation of and entrance fee to urban forests.

## 3. Results

### 3.1. Respondent Demographic Profiles

This research involved 254 respondents who were randomly selected among the visitors of Taman Tugu. The results of the respondent demographic survey (Table 2) show that most of the respondents were female (156 individuals (61.4%)), while 98 individuals (38.6%) were male, and 81.1% of respondents were Malays, followed by Chinese (10.2%), Indians (3.9%), and others (4.7%). Most of the respondents were aged between 41 and 50 years (30.7%), followed by respondents aged between 31 and 40 years (28.7%) and between 20 and 30 years (26.8%). Further, the educational level of the respondents showed that the majority had a bachelor's degree and had completed higher education (72.4%), followed by holders of a professional certificate (17.7%) and those who had completed high school (8.7%), while the remaining respondents were still in primary school. Looking at their employment status, 73.6% of respondents were working full-time, while 3.5% worked part-time, and 11% did not work. The highest income range for respondents was between MYR 5001 and MYR 10,000 (31.5%), followed by MYR 3001 to MYR 5000 (18.5%), and the remaining (11%) respondents earned less than MYR 500. It could also be seen that the majority (92%) of respondents lived in the Klang Valley area, while 8% lived outside that area; most of them (52%) were first-time visitors, while 27.6% had visited between two and five times, and about 20.5% had visited Taman Tugu more than five times. The majority of respondents (85%) had used their own vehicle as the means of transportation to reach the research area, followed by 13% who had chosen to carpool with friends, and the remaining had used either public transport, e-hailing, or office vehicles. Table 2 also shows that 57.9% of respondents had visited the study area together with their family members and 36.2% with friends, and the remaining had attended either alone, through a travel agency, or with office colleagues.

**Table 2.** Sociodemographic data of respondents.

| Demographic Background | Item | Frequency | Percentage (%) |
|---|---|---|---|
| Gender | Male | 98 | 38.6 |
| | **Female** | **156** | **61.4** |
| Age | <20 years old | 9 | 3.5 |
| | 20–30 years old | 68 | 26.8 |
| | 31–40 years old | 73 | 28.7 |
| | **41–50 years old** | **78** | **30.7** |
| | >51 years old | 26 | 10.2 |
| Race | **Malay** | **206** | **81.1** |
| | Chinese | 26 | 10.2 |
| | India | 10 | 3.9 |
| | Other | 12 | 4.7 |
| Education level | Primary school | 2 | 0.8 |
| | Secondary school | 22 | 8.7 |
| | Professional certificate | 45 | 17.7 |
| | **Degree and higher** | **184** | **72.4** |
| | Other | 1 | 0.4 |

**Table 2.** *Cont.*

| Demographic Background | Item | Frequency | Percentage (%) |
|---|---|---|---|
| Employment status | **Full-time worker** | **187** | **73.6** |
| | Part-time worker | 9 | 3.5 |
| | Unemployed | 28 | 11.0 |
| | Pensioner | 5 | 2.0 |
| | Student | 25 | 9.8 |
| Gross monthly income (MYR) | <500 | 28 | 11.0 |
| | 501–1500 | 12 | 4.7 |
| | 1501–3000 | 44 | 17.3 |
| | 3001–5000 | 47 | 18.5 |
| | **5001–10,000** | **80** | **31.5** |
| | >10,000 | 43 | 16.9 |
| Place of living | **Klang Valley** | **234** | **92.1** |
| | Outside Klang Valley | 14 | 5.5 |
| | Other | 6 | 2.4 |
| Frequency of visit | **1st time** | **132** | **52.0** |
| | 2 to 5 times | 70 | 27.6 |
| | >5 times | 52 | 20.5 |
| Transportation to the venue | Public transport | 4 | 1.6 |
| | **Own transport** | **216** | **85.0** |
| | Carpool | 33 | 13.0 |
| | Office vehicle | 1 | 0.4 |
| Company at the venue | Alone | 10 | 3.9 |
| | **With family** | **147** | **57.9** |
| | With friends | 92 | 36.2 |
| | With tourist agency | 1 | 0.4 |
| | Office colleagues | 4 | 1.6 |

*3.2. Visitor Perception of Urban Forests*

3.2.1. Normality of the Data

Normality assessment in this study was made by evaluating the skewness and kurtosis of each variable. The data distribution is declared normal if the tendency value is in the range of $\pm 2$, while the kurtosis index value must not exceed 10.0 [46]. Based on Table 3, the values of skewness were in the range of $-1.753$ to 0.88, and kurtosis had values between $-0.756$ and 2.412, thus not more than 10.0. This shows that the data obtained in this study had a normal distribution.

**Table 3.** Data normality.

| Variable | Skewness | Kurtosis |
|---|---|---|
| Level of knowledge of related terms | $-0.47$ | $-0.565$ |
| Perception of regulating services | $-1.753$ | 3.096 |
| Perception of provisioning services | $-0.28$ | $-0.756$ |
| Perception of cultural services | $-1.234$ | 0.771 |
| Perception of supporting services | $-1.704$ | 2.412 |
| Perception of urban forest amenities | $-0.798$ | 0.543 |
| Perception of issues created by urban forests | 0.880 | 0.470 |
| Perception of issues related to urban forest management | $-0.414$ | 0.489 |
| Perception of interest and trust in urban forest management | $-0.092$ | $-0.260$ |

3.2.2. Knowledge of Terms

A descriptive analysis involving mean and standard deviation was conducted to determine the knowledge of respondents of relevant terms. Table 4 shows the results. The majority of the respondents had a moderately high level of knowledge of the terms, with an

average score of 3.95, ±0.82. Out of the total score of 5, item number 2 showed the highest mean score, 4.51, indicating that the respondents had a high level of knowledge of the term 'ecosystem'; on the other hand, the score of the knowledge of the term 'urban' was 3.77, and the lowest mean score was that of the knowledge of the term 'ecosystem services' (3.59), although both items still scored moderately high.

**Table 4.** Knowledge of terms.

| No. | Item | Mean | SD | Interpretation |
|---|---|---|---|---|
| 1. | Term 'urban forest' | 3.77 | 1.205 | Moderately high |
| 2. | Term 'ecosystem' | 4.51 | 0.726 | High |
| 3. | Term 'ecosystem services' | 3.59 | 1.278 | Moderately high |
| | Total | 3.958 | 0.82238 | Moderately high |

### 3.2.3. Regulating Services

The results in Table 5 show that the majority of respondents had a positive perception (high-level scores of 4.74, ±0.40) of the regulatory services provided by urban forests. Improving air quality showed the highest mean score (4.85, ±0.38), while reducing noise pollution received the lowest score (4.65, ±0.62).

**Table 5.** Regulating services.

| No. | Item | Mean | SD | Interpretation |
|---|---|---|---|---|
| 1. | Reducing floods and landslides | 4.69 | 0.59 | High |
| 2. | Reducing heat in the city | 4.82 | 0.44 | High |
| 3. | Reducing noise pollution | 4.65 | 0.62 | High |
| 4. | Improving air quality | 4.85 | 0.38 | High |
| 5. | Helping the process of pollination or seed dispersal | 4.70 | 0.57 | High |
| | Total | 4.74 | 0.40 | High |

### 3.2.4. Provisioning Services

The results in Figure 3 show that most of the respondents had a positive perception of the provisioning services provided by urban forests (moderately high-level scores of 3.49, ±1.12). Item No. 1 received the highest mean score, 3.91, indicating that the visitors believed that urban forests provided drink resources, while item No. 3 had the lowest mean score, 3.15, indicating that the visitors did not believe that urban forests provided wood products.

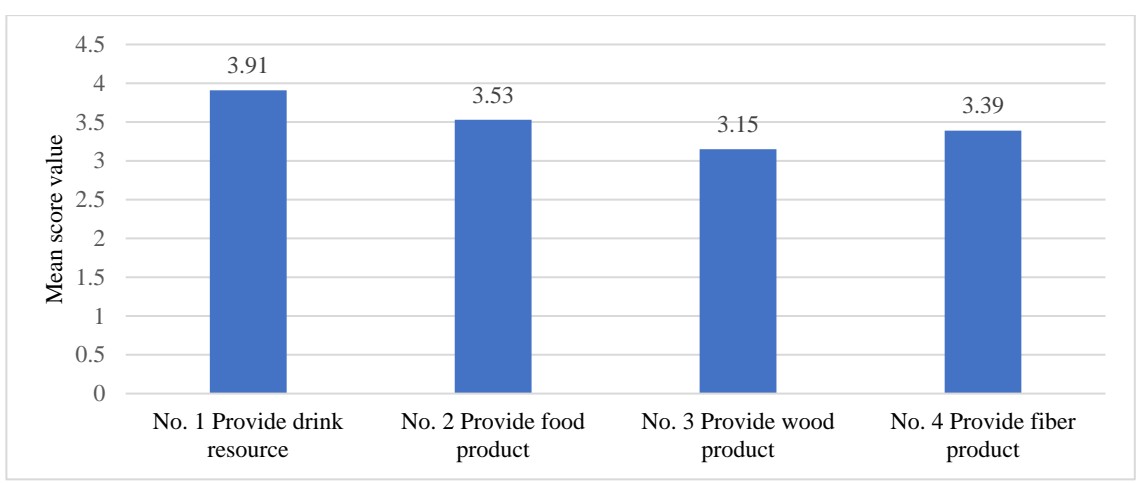

**Figure 3.** Provisioning services.

### 3.2.5. Cultural Services

Table 6 shows that the results of the study indicate that most of the respondents had a positive perception (high-level scores of 4.69, ±0.37) of the cultural services provided by urban forests. Item No. 3, i.e., providing beautiful scenery, received the highest mean score, 4.86, indicating that the visitors appreciated the beautiful view provided by urban forests.

**Table 6.** Cultural services.

| No. | Item | Mean | SD | Interpretation |
|-----|------|------|-----|----------------|
| 1. | Providing area for leisure and recreation activities | 4.83 | 0.43 | High |
| 2. | Providing area for education and learning purposes | 4.76 | 0.52 | High |
| 3. | Providing beautiful scenery | 4.86 | 0.37 | High |
| 4. | Providing peace and well-being | 4.85 | 0.39 | High |
| 5. | Increasing the value of the area | 4.20 | 0.99 | High |
| | Total | 4.69 | 0.37 | High |

### 3.2.6. Supporting Services

The results show that most of the respondents had a positive perception of the supporting services provided by urban forests (high-level scores of 4.70, ±0.50). Visitors believed that urban forests provided a habitat for flora and fauna (mean score of 4.74), while preserving genetic diversity scored 4.66.

### 3.2.7. Urban Forest Amenities

Table 7 shows that the results of the study indicate that most of the respondents had a positive perception of the facilities provided at the urban park, specifically, Taman Tugu (high-level scores of 4.39, ±0.53). Cleanliness scored the highest mean, 4.58, while the activities provided scored the lowest mean, 4.18, indicating that the visitors were satisfied with the facilities provided at the urban park.

**Table 7.** Urban forest amenities.

| No. | Item | Mean | SD | Interpretation |
|-----|------|------|-----|----------------|
| 1. | Public facilities (toilets, parking lot, etc.) | 4.34 | 0.75 | High |
| 2. | Information and education center | 4.26 | 0.74 | High |
| 3. | Pedestrian trail | 4.48 | 0.65 | High |
| 4. | Cleanliness | 4.58 | 0.60 | High |
| 5. | Safety | 4.50 | 0.62 | High |
| 6. | Activities provided | 4.18 | 0.78 | High |
| 7. | Management and maintenance | 4.43 | 0.66 | High |
| | Total | 4.39 | 0.53 | High |

### 3.2.8. Urban Forest Disservices

The results in Figure 4 show that most of the respondents had average and negative perceptions of the disservices of urban forests (total low-level mean scores of 2.19, ±0.95). The highest mean was that of urban trees causing brown waste, with 2.48, and the lowest mean was that of urban trees causing an unpleasant view, with 1.78, which indicates that urban forests did not cause a big issue for visitors.

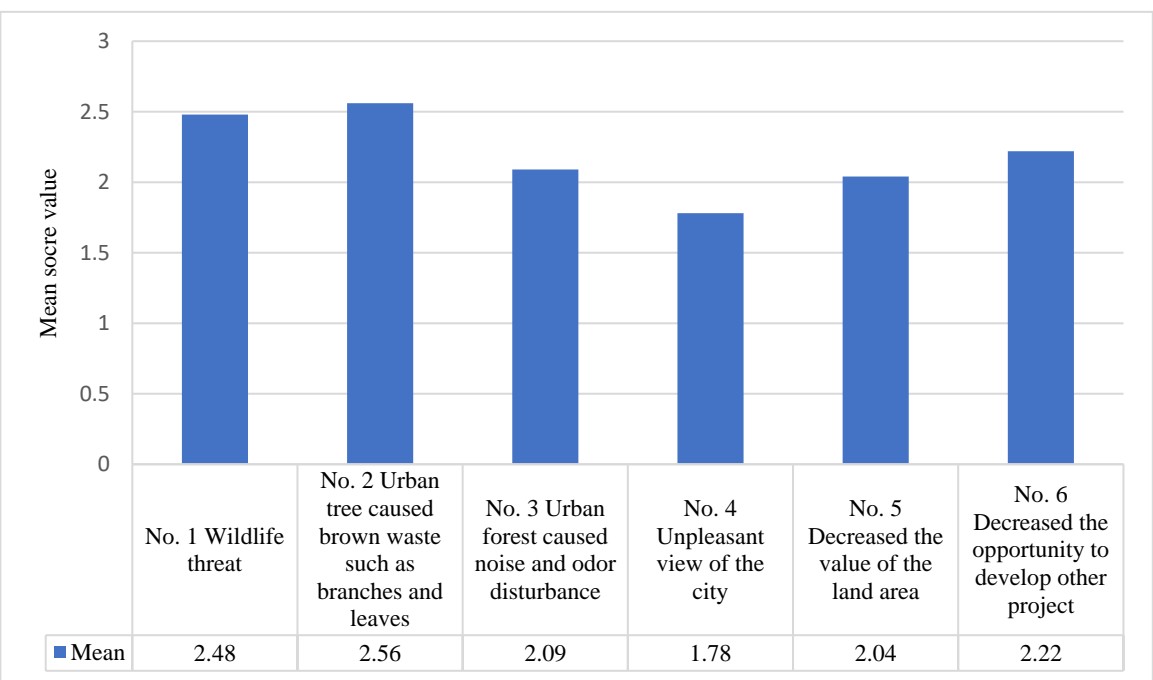

**Figure 4.** Issues caused by urban forests.

3.2.9. Urban Forest Management

The results show that the mean score of each item was moderately high or high. Items that had high scores were 'Lack of awareness among the public regarding urban forest ecosystem services' (4.07, ±1.01) and 'Lack of attention given by policy makers and decision makers in considering the importance of urban forests' (4.15, ±0.97), while the items that had moderately high scores were 'Lack of allocation for the purpose of preserving and conserving urban forests' (3.85, ±1.09), 'High management and maintenance costs' (3.54, ±1.12), and 'Less studies are conducted related to urban forests' (4.15, ±0.97). Overall, the mean scores of issues related to urban forest management were moderately high (3.9078, ±0.6118). Further, the mean score of 3.30, ±1.254 of the item 'Management and maintenance of urban forests should be fully handed over to the municipal council' was moderately high, while the item 'Management and maintenance of urban forests should be jointly managed by the government, private and public bodies' had a high score (4.47, ±0.823).

*3.3. Willingness to Pay for Urban Forests*

The results of the analysis showed that 79.1% said 'yes' to contributing a sum of money for conservation purposes, while the rest (20.9%) said 'no', and the average contribution proposed was as much as MYR 51.32 per year. For the second question related to WTP, 65% of respondents said 'yes', which means that they would agree to pay an entrance fee, with the average of minimum fee proposed being MYR 3.07 per entrée, while the remaining 35% said 'no'. These results indicate that the majority of the respondents were willing to pay and contribute to the conservation and management of urban forests.

Factors That Influenced Willingness to Pay

The model summary in Table 8 shows an R-squared value of 0.065, which means that only 6.5% of the variation in willingness to pay entry fees could be explained by variations in the twelve dependent variables. The balance (100% − 6.5% = 93.5%) was explained by other factors.

**Table 8.** Model summary.

| Model | R | R-Squared | Adjusted R-Squared | Std. Error of the Estimate | Change Statistics | | | | |
|---|---|---|---|---|---|---|---|---|---|
| | | | | | R-Squared Change | F Change | df1 | df2 | Sig. F Change |
| 1 | 0.255 [a] | 0.065 | 0.018 | 0.34489 | 0.065 | 1.391 | 12 | 241 | 0.170 |

[a.] Predictors: (constant), age, race, gender, income, education, employment status, company at the park, residence, frequency, transportation, ecosystem services, and amenities.

Based on Table 9, using the ANOVA test or F-test, the F-value obtained was 1.391, with a significance value of 0.170. Since the significance value was greater than 0.05, it could be concluded that the regression coefficients of the factors of age, race, gender, income, education, employment status, company at the park, place of residence, frequency, transportation, ecosystem services, and facilities did not affect the willingness to pay (WTP) for conservation purposes or entrance to urban forests.

**Table 9.** ANOVA test.

| | Model | Sum of Squares | df | Mean Square | F | Sig. |
|---|---|---|---|---|---|---|
| | Regression | 1.986 | 12 | 0.166 | 1.391 | 0.170 [b] |
| 1 | Residual | 28.667 | 241 | 0.119 | | |
| | Total | 30.654 | 254 | | | |

[b.] Predictors: (constant), age, race, gender, income, education, employment status, company at the park, residence, frequency, transportation, ecosystem services, and amenities.

Referring to Table 10, among the twelve variables included in the model, the only significant variable with $\alpha = 0.05$ was the perception of urban forest facilities ($\alpha = 0.045$). The other variables produced significant values greater than 0.05. This multiple regression analysis is used to test whether the independent variable can significantly predict the independent variable. The results of the regression analysis show that all the independent variables could significantly explain as much as 6.5% of the independent variables ($R^2 = 0.065$, F (2.254 = 1.391; $p > 0.005$)), while only urban forest facilities showed a negative and significant influence ($\beta = -0.139$, $p = 0.045$).

**Table 10.** Regression analysis.

| | Model | Unstandardized Coefficients | | Standardized Coefficients | t | Sig. |
|---|---|---|---|---|---|---|
| | | B | Std. Error | Beta | | |
| | (Constant) | 1.920 | 0.340 | | 5.648 | 0.000 |
| | Gender | −0.091 | 0.049 | −0.128 | −1.861 | 0.064 |
| | Age | −0.010 | 0.026 | −0.029 | −0.377 | 0.707 |
| | Race | 0.037 | 0.029 | 0.084 | 1.265 | 0.207 |
| | Education status | −0.012 | 0.036 | −0.023 | −0.334 | 0.739 |
| | Employment status | −0.019 | 0.022 | −0.071 | −0.844 | 0.400 |
| 1 | Monthly income | −0.028 | 0.023 | −0.125 | −1.256 | 0.210 |
| | Place of living | 0.057 | 0.060 | 0.061 | 0.946 | 0.345 |
| | Frequency of visit | 0.040 | 0.029 | 0.091 | 1.362 | 0.175 |
| | Means of transportation | 0.025 | 0.058 | 0.029 | 0.429 | 0.669 |
| | Company at the park | 0.013 | 0.036 | 0.025 | 0.377 | 0.707 |
| | Ecosystem services | −0.031 | 0.056 | −0.039 | −0.564 | 0.573 |
| | Amenities | −0.090 | 0.045 | −0.139 | −2.011 | 0.045 |

## 4. Discussion

### 4.1. Visitor Perception of Urban Forests

The results of the study show that visitors gave high scores to the level of ecosystem services provided at Taman Tugu. This shows that the ecosystem services of the Taman Tugu urban forest generated a positive perception in visitors. Regulatory services had the highest average mean score (mean = 4.74). Visitors believed that urban forests can reduce floods and landslides, reduce heat in urban areas, reduce noise pollution, improve air quality, and help pollination and seed dispersal. There are many studies that have proven that urban forests play an important role in improving air quality [7], reducing noise pollution [8], and reducing temperatures in urban areas [6]. Cultural services also generated a positive perception (mean = 4.69), and visitors believed that the Taman Tugu urban forest provides facilities for leisure and recreation, and for learning and education, as well as beautiful scenery, peace, and well-being. Basu and Nagendra [47], in their study, stated that apart from performing physical activities, urban parks are also places to carry out recreational activities, observe nature, and conduct activities that play an important role in maintaining health and mental well-being. A study conducted by Chiesura [48] in Vondelpark, Amsterdam, also found that most users visited the park to rest. In addition, visitors of Taman Tugu also agreed that urban forests increase the value of real estate in the surrounding area. This was also proven by studies conducted by Georgi and Dimitriou [49], who stated that the value of property in residential areas and commercial areas close to parks and open spaces increases.

The results of the study also show that this urban forest also provides supporting services, with a high average score (mean = 4.70), by providing habitat for flora and fauna and at the same time preserving genetic diversity. Urban green areas are biodiversity hotspots for various species of plants and animals [12] and various species of birds and bees that play an important role in pollination and seed dispersal [50,51]. Shackleton et al. [52] also stated that supporting services are services that support the balance of biodiversity and are an important basis for other services, in addition to supporting the activities carried out by humans. Regarding provisioning services, the score obtained was moderately high (mean = 3.49) compared with other services. This is because ecosystem services in urban forests do not provide direct benefits to the local community, such as supplying water sources for drinking or supplying products such as food, wood, and fiber. Urban forests usually have a small area and are not seen to function as forests that supply products or needs to the local community. Sources of water, food, wood products, and medicine are obtained from reserve forests, which have a larger area and are rich in ecosystem services. Although the role of provisioning services in the urban forest ecosystem is seen as small, it is still important in ecosystem balance and has potential in global food security [53]. Recent studies have also found that urban ecosystems also play a role in supplying food and other products to urban communities [54–56]; in addition to the potential of community gardens and urban gardens, they serve as places to share knowledge related to bio culture [57].

Facilities such as fields [58], trails, or paths [59], as well as the whole facilities, play a role in promoting the use of parks and physical activity. Overall, respondents had a positive perception, with a high score (mean = 4.26), of the amenities provided at Taman Tugu, including the aspects of public facilities such as toilets, parking spaces, educational information centers, signage, and pedestrian walkways. Visitors were also satisfied with the cleanliness of the area, and the safety and the activities provided. Management and maintenance at Taman Tugu also received a high score. This shows that the Taman Tugu urban forest is well managed and maintained and highly satisfies most visitors. Safety is one of the main factors that determine the number of urban green areas [60–62]. Although there is a study that shows that the urban community believes that urban forests cause problems such as safety issues and the threat of wildlife [63], this study found that visitors at the Taman Tugu urban forest gave positive scores to the level of safety at Taman Tugu (mean = 4.50). This is also in line with the findings of a study conducted by Sreetheran [64], where 80% of visitors felt safe when they were in urban parks in Kuala Lumpur. Among

the factors that cause visitors to feel safe while in the Taman Tugu area may be the fact that the park has security guards on duty throughout the operating hours to ensure the safety of visitors; further, the public is not allowed to enter the park area after 6.30 pm for safety reasons and to avoid unwanted incidents. Urban forest disservices such as the threat of wildlife, brown waste from leaves and tree branches, noise and smell disturbances, and visual disturbances obtained a moderately low average score (mean = 2.19). This shows that visitors do not perceive those elements as a threat when carrying out activities in Taman Tugu.

### 4.2. Willingness to Pay for Urban Forests

The findings of this study showed that most respondents (79.1%) were willing to pay, for urban forest conservation, an average payment of MYR 51.32 per year, while 20.9% were not willing to pay. A previous study by Rahman et al. [65] also found that 60% of residents were willing to pay for the conservation of Bukit Nanas Forest Reserve, Kuala Lumpur, suggesting conservation fees of MYR 2 to MYR 5 per year, while 40% of residents were not willing to pay. Similarly, the study conducted by Samdin et al. [66] reported that most Penang National Park visitors expressed their willingness to make a contribution for conservation purposes, with an average WTP value of MYR 6.30 for local visitors and MYR 9.63 for international visitors. Next, a study conducted by Mamat et al. [67] found that the average rate of WTP for conservation purposes on Redang Island was as much as MYR 16.87, while the study conducted by Kamri [68] in Gunung Gading National Park in Sarawak showed that the average WTP rate was MYR 10.63. Sharip and Abdul Ghani [69], in their study, also found that 62% of respondents were willing to pay for lake conservation, with average WTP rates of MYR 13.71 for Lake Putrajaya and MYR 14.35 for Lake Muda, Kedah; further, the findings obtained by Rosli et al. [70] indicate an average WTP rate of as much as MYR 12.93 for the conservation of fireflies in Kuala Selangor. Kuala Lumpur City Council [42] stated that the average entrance fees to enter the urban forests in Kuala Lumpur range from MYR 5.00 to MYR 30.00 depending on the area and its amenities. For example, the Bukit Seputeh urban forest charges visitors around MYR 5.00 (children) and MYR 10.00 (adults). On the other hand, KL Forest Eco Park charges visitors MYR 20.00 for children and MYR 30.00 for adults. This shows that the entrance fees charged in Kuala Lumpur revolve around the services provided by the urban forest. Bukit Seputeh only provides spiritual (cemetery area) and few regulating services; however, KL Forest Eco Park not only provides regulating services but also aesthetic services, recreational services, educational services, etc. That is why the entrance fee for KL Forest Eco Park is higher than that for the Bukit Seputeh urban forest.

Based on the research conducted, there are some significant differences between the average WTP value suggested by the visitors of Taman Tugu and the WTP value suggested by visitors in several natural attraction locations in Malaysia. This may be due to several factors. First, each study location has different attractions that cause visitors to make an assessment based on the experience gained during their stay at that location and further causes the value of the ecosystem services provided to differ. The background of visitors may also be different, considering that the location of this study was a metropolitan urban area and most visitors were urban residents who had high-level income and education and thus were willing to pay a higher rate to enjoy the services provided by urban forests. The value of services perceived by individuals may also vary even when considering the same gender and age, because everyone has different interests and tendencies. This could also cause the ringgit value they are willing to pay to be different [71]. Furthermore, the WTP method used also contributes to the difference in the average WTP value. A study conducted by Ahmed and Gotoh [72] showed that the use of either the open-ended or dichotomous choice method has its own advantages and disadvantages. Studies carried out by previous researchers used different methods for determining WTP; for example, Sharip and Abdul Ghani [69] and Abdul Aziz [73] used the bidding method, where the WTP value needs to be set within a certain range to be chosen by respondents, while

Rosli et al. [70] used the double-bounded, dichotomous choice method to obtain the WTP value. However, in this study, where the researchers did not require any amount to be bid, respondents indicated the maximum value they were willing to pay for urban forest conservation. This study also found out that only 35% were not willing to pay urban forest entrance fees, while 65% were willing to pay an average rate of MYR 3.07 per person. This value is slightly lower than that found in the study conducted by Siew et al. [74] in Paya Indah Wetland, which was as much as MYR 7.12 per person. However, a study conducted by Abdul Aziz [73] in Putrajaya Botanical Garden and Bukit Nanas Forest Reserve found that most respondents were willing to pay MYR 0.50 per person, while the average WTP value according to visitors at Gunung Stong State Park, Kelantan, was MYR 5.03. Thus, the WTP rate that visitors were willing to pay in Taman Tugu is not much different from the rate obtained in other studies. This review did not place any value on bids. Therefore, respondents were free to indicate the value that they felt was an appropriate entrance fee for the Taman Tugu urban forest.

*4.3. Factors Influencing Willingness to Pay*

This study further aimed to statistically identify factors that potentially influence the willingness to pay for urban forest conservation and entrance. Based on this study, it was found that the sociodemographic factors that were studied (gender, age, race, education level, employment status, income, frequency of visiting, means of transportation, places of interest, and company at the park) did not show any significant relationship with WTP, with significance values greater than 0.05. This also includes the visitor perception of the ecosystem services provided by urban forests, with a significance value $\alpha = 0.573$. Only the perception of urban forest amenities had a significant relationship, with $\alpha$ value $= 0.045$. Referring to the results of the descriptive analysis carried out on urban forest facilities, visitors had a positive perception of urban forest amenities (mean $= 4.39$, SD $= 0.53$). Thus, this study found that a positive perception of amenities such as public toilets, signage, parking lot, recreational areas, and other facilities provided is a factor that contributes to the willingness to pay for either urban forest conservation or entrance. Similar findings were also obtained by Tian et al. [51], according to whom, perception plays a role in WTP. Respondents who had a positive perception of the environment were willing to pay more for conservation purposes. This is also in line with the findings of the studies by Abas et al. [50] and Rosli et al. [70]. According to Ashaari and Johari [75], the more positive the feelings of individuals are, the higher their sympathy for fauna and support for conservation efforts are. The study by Song et al. [76] also found that visitors do not want to pay if the park and facilities provided are not well maintained and bring harm to visitors.

Next, the results of the study found that only 20.9% of respondents were not willing to pay for urban forest conservation and 35% were not willing to pay an entrance fee. There were several factors that caused respondents to be unwilling to make any contribution. The main factor was that they believed that urban forest conservation is under the responsibility of the government and local authorities; in addition, they were afraid that the donation given would be misused. Taxes paid to the government should be used and allocated to forest conservation, including urban forests. The main reasons found here were similar to those found by López-Mosquera et al. [77] and Tian et al. [51]; in their studies, the respondents who opted not to pay for urban green space conservation felt that they paid enough tax or expressed concerns about how their money would be used [78]. In addition, it was mentioned that the government should play a social role in the provision and maintenance of urban forest facilities for the benefit of the dwellers. Therefore, the government needs to provide funds and allocations for the purposes of management, preservation, and conservation of urban forests. This is in line with the research conducted by Othman and Jafari [79], which indicates that urban park services should be provided free of charge because public facilities and their funding are under the responsibility of the government. Respondents also stated that an urban forest park is a natural park developed for public use. Therefore, it is not appropriate for an entrance fee to be charged to visitors. Charging

an entry fee could also cause a decrease in the rate of visitors, reduce the motivation of visitors to carry out leisure activities, and cause visitors to switch to parks that provide facilities for free. They also mentioned that providing facilities to the public in urban forests should become a governmental initiative for ensuring balance in life and providing a place for recreation, a platform to spread awareness in the community about the importance of forests, and a place for the young generation to learn and appreciate the environment. Some of the respondents also mentioned that they were not willing to contribute due to financial constraints, and this is also in line with the study conducted by Nordin et al. [80], where students and low-income groups experienced difficulty in making payments. Furthermore, the effect of the COVID-19 pandemic on the economic status of Malaysians in general and Kuala Lumpur residents in particular was identified as another factor. The movement control order implemented by the government caused the economy to be paralyzed and had an impact on jobs, income, and lifestyle, disrupting the supply chain and business and worsening the situation of inequality, poverty, and hardship, especially for the most vulnerable groups [81].

## 5. Conclusions

Urban forests play an important role in ensuring sustainable and livable urban development. The ecosystem function of a city is incomplete if the management of urban forests is not upgraded and improved, since urban forests play an important role as a mitigator in solving the issues related to the well-being of the population and the environmental issues that occur in the city. Based on this study, there are a number of recommendations that can be made. In this study, we found that the majority of community members are willing to make contributions towards the conservation of and entry to urban forests, as well as assigning a monetary value to the ecosystem services provided. This is due to their appreciation of urban forests, awareness of the benefits derived from the environment, and the contributions to the well-being of urban residents. They are also aware that a certain amount of financial expenditure is needed to preserve, conserve, and maintain urban forests. Therefore, comprehensive financial planning and management are required.

This study also found that WTP is significantly related to the perception of the amenities provided by urban forests. In this case, the management teams of forests need to carry out regular maintenance to ensure that the urban forest facilities are at a good level, that the facilities provided are improved over time, and that the safety of the area is guaranteed to ensure the satisfaction of visitors when visiting and carrying out activities in urban forests. At the time this study was carried out, the management and maintenance of Taman Tugu urban forest were handled by a non-governmental body. Thus, governmental agencies and local authorities can use the concept, management, and maintenance of this park as a reference and an example to be used and applied to attract more people and encourage them to visit, perform recreational activities, and enjoy the facilities provided and at the same time, to create awareness about the environment. This could also simultaneously improve the quality of life of urban communities in Malaysia. On the other hand, a comprehensive policy needs to be formulated to support all these actions. Finally, it is hoped that all parties involved can plan, develop, and manage urban forests by holistically encompassing all aspects, including taking into account the value of ecosystem services provided by urban forests, providing financial allocations for conserving and managing urban forests, being transparent and accountable, making fair and equitable access to resources possible, and encouraging the involvement of the public in making any plans and decisions to ensure that the sustainable development goals can be achieved.

**Author Contributions:** Data curation, E.S.J.; Formal analysis, A.A.; Investigation, E.S.J.; Methodology, A.A.; Resources, A.A.; Supervision, A.A.; Validation, E.S.J.; Visualization, A.A.; Writing—original draft, E.S.J.; Writing—review & editing, E.S.J. & A.A. All authors have read and agreed to the published version of the manuscript.

**Funding:** This research was funded by Universiti Kebangsaan Malaysia through research grant SK-2022-015.

**Data Availability Statement:** Not available.

**Acknowledgments:** Many thanks to Universiti Kebangsaan Malaysia for providing facilitation to complete this study.

**Conflicts of Interest:** No potential conflict of interest was reported by the authors.

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
