# Peer review of "Valuation of Visitor Perception of Urban Forest Ecosystem Services in Kuala Lumpur"

_land, doi:10.3390/land12030572_

Round 1

Reviewer 1 Report (Previous Reviewer 1)

Thank you.

This is a well-revised article.

I can't find any more critical points

Reviewer 2 Report (Previous Reviewer 2)

The questions or uncertainties I commented after reviewing this paper would be of significance for readers and the persons in charge of park management administration.  I am sorry that the authors didn't reply to them.

Reviewer 3 Report (Previous Reviewer 3)

Dear colleagues, I will not repeat myself, my opinion about the article remains the same. I agree with my fellow reviewers, with their comments. Probably, from the standpoint of sociology, there are questions to work. But I evaluate it from an ecological standpoint. I did not find direct answers to my comments in the new version of the article. However, they were advisory in nature, so I do not insist on taking them into account.

This manuscript is a resubmission of an earlier submission. The following is a list of the peer review reports and author responses from that submission.

Round 1

Reviewer 1 Report

1. Map, Ecosystem

Even though the author focused on the conservation value of urban forests, the author needed to show the detailed area map and ecosystem (species of plants, trees, wildlife) and photos of the Taman Tugu.

(The author wrote in several sentences, but the reader won't know the shape of that species and landscape)

We can see just a vague land map of Kuala Lumpur, but the reader wants to know precisely the target area's current status.

Please add photos of urban forests and a detailed map (GIS data, Satellite map).

2. WTP method

WTP is a good indicator. I know much previous research on the WTP method in parks, national parks, and ecotourism areas. Still, as we know, there is much criticism about WTP, too (ex., Respondents pretend to have a positive attitude toward nature conservation). I’d like to suggest adding some interpretation in the manuscript with this critical point on WTP; thus, the author’s interpretation will be more balanced, but the author already recognized the weak point, and well explained. Then I wonder why the author uses this method directly; even the author knows well the weak point.

I suggest the author add more explanation about the Entry fee to Urban forests in Kula Lumpur because the reader needs to learn about the entry fee in Malaysia.

It’ll be better if the author can show a comparable price—for example, the Entry fee for Zoo, festivals, concerts, and exhibitions in Kuala Lumpur, Mega City, not just a conservation area in Malaysia. Then the reader can better estimate the perception of urban dwellers toward the urban forest in Kuala Lumpur.

The author just showed an example of WTP results in a national park or protected area of a non-urban site. 

Reviewer 2 Report

In this paper, a perception survey was conducted to grasp the people's basic knowledge and value recognition of ecosystem services of urban forests. In addition, the survey includes about the entrance fees as beneficiary liability for the conservation and management of urban forests.

I could not judge whether this questionnaire survey was appropriate or not, because the specific contents of the questionnaire were not described. For example, it is unclear how many questions were asked about the technical terms (basic knowledge, regulating services, cultural services and supporting services) and what the expected answers were.

And the questionnaire results were scored in this paper. However, there were not any explanations on what criteria the answers to the questions are scored. Especially for descriptive questions, it is likely that the evaluators conducting this survey may have different criteria.

The survey target is urban forest. And the survey respondents were limited only to the visitors of the forest in this paper. However, in order to discuss the significance of the existence of urban forests, the beneficiaries should include not only visitors but also the surrounding residents of urban forests. Considering funding for the conservation and maintenance of forest ecosystem services, it should broadly include people other than visitors. Otherwise, it will be difficult to make appropriate administrative decisions on beneficiary liability.

In this sense, unfortunately, I cannot positively evaluate this paper.

Reviewer 3 Report

The problem of preserving and assessing the importance of urban green spaces is of an international nature. All regions of the world face this problem. The importance of its effective solution is increasing every year. The concept of ecosystem services, including those provided by urban ecosystems, is well known to specialists and has been developed in sufficient detail. It is this concept that makes it possible to evaluate the ecosystem services of urban ecosystems. However, the main purpose of this article is to assess the perception of ecosystem services by park visitors of different races, ages, income levels, education, etc. This kind of research is very interesting because it makes it possible to find out the level of people's understanding of ecosystem services and evaluate it, including in economic indicators (willingness to pay for ecosystem services). This is the relevance and significance of this work. The methodology of the work is presented in detail, a comparison with the results of similar studies is carried out, all references are relevant. The results obtained are generally logical and correspond to the tasks set. The main result is that the population understands and positively evaluates the ecosystem services of urban forests. The majority of the population is ready to pay for the preservation of forests and ecosystem services, and priority services for the population have been identified (the contribution of forests to ensuring clean air, aesthetic appeal, etc.). The fears of people who do not want to pay for the preservation of forests are also understandable – they are international and characteristic of large cities (city authorities are responsible for urban green spaces, assume the possibility of using funds for other purposes, etc.). In general, these are very serious results that give grounds for making decisions in the field of urban green areas management.

The article is presented in sufficient detail and professionally, so there are no significant proposals for its improvement. Probably, the need for such a detailed socio-demographic analysis of respondents should have been explained more clearly (Section 3.1., Table 2). Since further differences in the perception of ecosystem services by different groups of respondents (gender, race, age, income, etc.) are not considered, and this analysis is of an independent nature. The differences between the groups are given only by willingness to pay for ecosystem services (Table 10). It can also be noted that you should not duplicate the words from the title of the article in the "keywords" section,